# Biodiversity of Herbivores Triggers Species Differentiation of Coprophilous Fungi: A Case Study of Snow Inkcap (*Coprinopsis* sect. *Niveae*)

**DOI:** 10.3390/jof10120835

**Published:** 2024-12-02

**Authors:** Liyang Zhu, Tolgor Bau

**Affiliations:** 1Engineering Research Centre of Chinese Ministry of Education for Edible and Medicinal Fungi, Jilin Agricultural University, Changchun 130118, China; zly2003@126.com; 2Key Laboratory of Edible Fungi Resources and Utilization (North), Ministry of Agriculture, Changchun 130118, China

**Keywords:** coprinoid fungi, coprophilous fungi, snowy inkcap, morphology, multigene phylogeny, taxonomy, new taxa

## Abstract

Coprophilous species of *Coprinopsis* sect. *Niveae*, commonly known as “snow inkcap”, are widespread in pastoral areas; however, wide sampling approaches are needed to discover new taxa and to clarify the taxonomic status of the so-called “snow inkcap”. Nationwide field work was conducted in China with a detailed record collected of the distribution and the animal origin of the dung. A four-loci phylogenetic study of *Coprinopsis* sect. *Niveae* was conducted based on the internal transcribed spacer regions (*ITS*), the ribosomal large subunit (*LSU*), and translation elongation factor 1-α (*tef1-α*)], and the mitochondria small ribosomal RNA subunit (*mtSSU*). Fourteen phylogenetic species were assigned to this section, including six novel species, namely *Coprinopsis furfuracea*, *C. iliensis*, *C. khorqinensis*, *C. sericivia*, *C. subigarashii*, and *C. tenuipes*. Macro-, micro-, and ultramicro-morphological observations of species collected from China were also conducted and the detailed descriptions and illustrations of the novel species are provided. Our studies revealed that the different origin of herbivore dung, the distribution, the color and thickness of the pileus, the shape of stipes, the shape and size of basidiospores, and the presence or absence of pleurocystidia can be used as characteristics for distinguishing species in sect. *Niveae*. The key to species belonging to this section is also provided.

## 1. Introduction

Inkcap mushroom, also known as coprinoid fungi, derives its name from the Greek “Kopros”, meaning “dung”, as they commonly grow on herbivore dung, although Fracastorius [1]. *Coprinopsis nivea* (Pers.) Redhead, Vilgalys & Moncalvo, also known as “snowy inkcap”, is one of the most famous coprophilous fungi, often found on cow and horse dung. *Coprinopsis nivea* and related species play essential roles in organic matter decomposition, nutrient cycling, and maintaining ecological balance in grasslands and forests. Due to their close association with herbivore feces, researchers have used their basidiospores as indicators of large herbivore presence and distribution, as well as archaeological evidence of human–animal husbandry practices [2,3,4].

The classification of *C. nivea* and related species has a complex history. Originally, Persoon [5] described it as *Agaricus niveus*, noting its snow-white campanulate pileus and farinaceous-squamulose stipes, which grows on dung under section *Coprinus*. This early classification set the foundation for subsequent studies on this group, which were grounded in the broader concept of *Coprinus* Per. [6,7,8].

Fries [9] placed this species in the *Comati* of Trib. *Pelliculosi* based on the fact that the center of its pileus is carnulose and ovoid, which is covered by white fluffy-squamous furfuraceous scales. He also mentioned that *C. niveus* is similar to other species in the Trib. *Veliformes* with a slender pileus and stipes, such as *Coprinus ephemeroides* (DC.) Fr. (now, as a member of *Narcissea* D. Wächt. & A. Melzer) and *Coprinus narcoticus* (Batsch) Fr., which is also attributed to *Coprinopsis* now in the same study.

Massee [6] divided *Coprinus* s.l. into six groups based on their macroscopic characteristics. He placed *C. niveus* in section IV, which comprised species with obvious but not glistening and micaceous veils, and believed that this species may be similar to *Coprinus tuberosus* [for example, *Coprinopsis tuberosa* (Quél.) Doveri, Granito & Lunghini], which has the same scaly powdery pileus and sclerotia.

Lange [10,11] divided *Coprinus* into *Comati*, *Nudi*, and *Farinosi* based on their macroscopic and microscopic characteristics. *Farinosi* was subdivided into *Annulati* and *Exannulati* based on the presence or absence of rings on the stipes. *Coprinus nivea* was attributed to *Vestiti* of *Exannulati*, which belongs to the group *Farinosi* because of its absence of a ring, and the presence of thick powdery and flocculent veils composed of (sub)globose cells. Lange observed that the veil elements of *C. niveus* were almost smooth; however, *Coprinus narcoticus*, another member of this group, had veil cells with obvious warts. Kühner and Romagnesi [12] further subdivided the *Vestiti* into subsection *Nivei* and the “stercorarius group” based on the differences in the presence or absence of warty ornamentation on their veil elements and episporium of basidiospores and the odor of the basidiocarp. They also thought that species, closely related to *C. niveus*, might include *Coprinus coniophorus* and *Coprinus cortinatus*, which are currently attributed to *Coprinopsis*, and *Coprinus ephemeroides* and *Coprinus patoullardii* as members of *Narcissea*. The divided classification system of the *Vestiti* was also approved by Kits van Waveren [13].

However, Orton and Watling [14] suggested that coprinoid fungi with globose veil elements should be assigned to sect. *Micaceus*, and designated *Coprinus niveus* within the stirps *Niveus,* in this section. They proposed adding *Coprinopsis cothurnata* (Godey) Redhead, Vilgalys & Moncalvo; *Coprinopsis pachysperma* (P.D. Orton) Redhead, Vilgalys & Moncalvo; *Coprinopsis poliomalla* (Romagn.) Doveri, Granito & Lunghini; and *Coprinus cordisporus* as members of stirps *Niveus*. Additionally, *Coprinopsis cortinata* (J.E. Lange) Gminder, due to its veil elements mixed with mycelial/fibrous cells, was attributed into stirps (*Cortinatus*). Citerin [15] followed the classification system proposed by Orton and Watling [14], and confirmed *Micaceus*, *Farinosi*, and *Niveus* as subgenus, sections and subsections, respectively.

Uljé and Noordeloos [16] identified *Coprinus pseudoniveus* as the second species belonging to this subsection. They maintained that it differed from *Coprinus cortinata*, a species in subsect. *Nivei* having a mealy–powdery veil and cortina veils at the top and edge of the pileus, respectively. Further, their veils were smooth or covered with crystals or granules that dissolved in HCl solution.

With the development of molecular phylogeny, the taxonomic status of *C. nivea* and related species has been stable. Based on the phylogenetic results of Hopple and Vilgalys [17,18], *Coprinus* s.l. has been formally divided into four genera: *Coprinus* Per.; *Coprinopsis* P. Karst.; *Coprinellus* P. Karst.; and *Parasola* Redhead, Vilgalys & Hopple by Redhead et al. [19]. *Coprinus niveus* should be placed in *Coprinopsis*, which is also confirmed by morphologic features such as the pileipellis in cutis type and the presence of the veil. However, the taxonomic status of small coprophilous species such as *Coprinus cordisporus* in the original subsect. *Nivei* has not yet been resolved and should be excluded from *Coprinopsis*. Schafer [20] adopted the framework of the four genera proposed by Redhead et al. [19], and established *Coprinopsis* sect. *Niveae*; however, the members included in this section were not clear in Schafer’s framework. Gierczyk et al. [21] considered this section (as sect. *Nivei*) including *C. nivea*; *C. pseudonivea* (Bender & Uljé) Redhead, Vilgalys & Moncalvo; *C. cortinata* (J.E. Lange) Gminder; *C. utrifer* (Joss. ex Watling) Redhead, Vilgalys & Moncalvo; *C. bellula* (Uljé) P. Roux & Eyssart.; *C. coniophora* (Romagn.) Redhead, Vilgalys & Moncalvo; *C. cothurnata* (Godey) Redhead, Vilgalys & Moncalvo; and *C. poliomalla* (Romagn.) Doveri, Granito & Lunghini based on ITS datasets. However, the long divergence between the latter two species renders this framework unstable.

Based on the phylogenetic results of Psathyrellaceae, Wächter and Melzer [22] considered the monophyly and divergence of the subgenus classification, and confirmed the members of sect. *Niveae*: *C. nivea* (the type species in this section); *C. pseudonivea*; *C. afronivea* Desjardin & B.A. Perry; *C. igarashii* Fukiharu & Kim. Shimizu; and *C. yonkinensis* (unpublished species). Moreover, *C. cortinata*; *C. bellula*; *C. utrifera* (Joss. ex Watling) Redhead, Vilgalys & Moncalvo; *C. cerkezii* Tkalčec, Mešić, I. Kušan & Matočec; and *C. coniophora* are as members of sect. *Subniveae* D. Wächt. & A. Melzer, sister to sect. *Niveae*; in addition, they established *Narcissea*, which included little fimetarius species with powdery veils, such as *Narcissea cordispora,* which are thought to be members of sect. *Nivei*. Currently, the scope of sect. *Niveae* remains controversial. On the one hand, although phylogenetic studies support the establishment of sect. *Niveae* and sect. *Subniveae*, there is no obvious difference in the description of morphological characteristics between these two sections. On the other hand, taxonomic status of some species that were once considered to be members of *Coprinus* sect. *Nivei* are currently undetermined. For example, our previous studies have shown that the so-called “*Coprinopsis empheroides*” should be included in *Narcissea* as a new combination and excluded from sect. *Niveae* [23]; however, this conclusion has not been widely recognized.

The classification system by Wächter and Melzer [22] is adopted. To date, three species have been identified in sect. *Niveae*, namely *C. nivea*, *C. pseudonivea*, and *C. afronivea* in China [23]. However, almost all the fecal species of this section were recognized as *C. nivea*, which might contain different species. In this study, we discuss the following: (1) the six new species of this section that were found in China, and we provide a key to sect. *Niveae*; (2) the description of sect. *Niveae*; and (3) the differences between sect. *Niveae* and sect. *Subniveae*.

## 2. Materials and Methods

### 2.1. Sampling and Morphological Characterization

Five years of fieldwork in China were conducted from 2019 to 2024, and 54 collections were collected in Heilongjiang Province, Jilin Province, Inner Mongolia Autonomous Region, Jiangsu Province, Hubei Province, Guangxi Autonomous Region, Yunnan Province, Xinjiang Uygur Autonomous Region, Qinghai Province, and Xizang Autonomous Region of China. Specimens were photographed, tagged, and ecological information recorded before being collected in the field. Macro-morphological descriptions of color were based on the Methuen Handbook of Colour [24]. The samples were dried at 45 °C for at least 8 h in an oven (Xiaoxiong, Foshan, China). After observation, voucher specimens were deposited in the Herbarium of Mycology at Jilin Agricultural University (HMJAU), if not otherwise indicated.

To prepare slides for the microscopic observation, the specimens were sectioned by hand under a stereoscope (stemi 2000C, Zeiss Co., Ltd., Oberkochen, Germany). Then, the slides were prepared in water and 5% aqueous KOH, and 1% Cango Red solution or 1% acetocarmine stain added if necessary. Under a light microscope (BX53, Olympus Co., Ltd., Tokyo, Japan), microscopic features with at least 40 structures for the new species and 20 for the known species were measured, including the size, shape, and color of basidiospores, basidia, pseudoparaphyses, cheilocystidia, pleurocystidia, pileocystidia, caulocystidia, pileipellis, stipipellis, gill trama, subpileipellis, and stipe trama. Additionally, the presence of clamp connection was observed in each sample. Free mature basidiospores collected from the spore print were chosen for observation in front view and/or side view through 1000× magnification and measurement by software EP viewer V1.4 (Olympus Co., Ltd., Tokyo, Japan) with precision to 0.01 μm; the diameter of the germ pore was also measured, and the hilum was excluded under any situation. Statistical results of measured values are presented as (a) b–c (d) form, b–c represent the 90% confidence interval, and (a) and (d) represent minimum and maximum values. Basidiospore sizes are presented as follows: length range × breadth range × width range. The Q value was calculated as Q = length divided by width; when breadth range was measured, Q1 = length divided by breadth range and Q2 = length divided by width [16]; the shape terms corresponding to the Q value were described according to Bas [25]. Other structures were measured and described through 400× magnification. In the case of basidia, the length of the sterigma was excluded from the length of the basidia, and the value of the widest part was chosen as the widest length of cystidia and basidia. The clusters of basidia length were used to classify the morphological types [26,27].

To examine the superficial characters of the basidiospores of species in sect. *Niveae*, scanning electron microscopy (SEM, Zeiss MERLIN, Jena, Germany) was conducted using the protocol of Zhou and Bau [28].

### 2.2. DNA Extraction, PCR Amplification and DNA Sequencing

The genomic DNA from each isolate was extracted from the herbarium voucher and amplified following the protocols of Mou and Bau [29]. For each species, we analyzed four loci: three loci located in the nucleus [the internal transcribed spacer regions (*ITS*), the ribosomal large subunit (*LSU*), and the translation elongation factor 1-α (*tef1-α*)] and one in the mitochondria [the mitochondria small ribosomal RNA subunit (*mtSSU*)]. The primer sequences used for amplifying the four loci were ITS1F/ITS4 for *ITS* [30], LR0R/LR7 for *LSU* [31], 983F/2218R for *tef1-α* [32], and MS1/MS2 for *mtSSU* [30]. The DNA sequencing was conducted by Sangon Biotech Co., Ltd. (Shanghai, China). All newly generated sequences were deposited in GenBank (www.ncbi.nlm.nih.gov/genbank, accessed on 25 November 2024), and all sequences used for phylogenetic studies are listed in Table 1.

### 2.3. Alignment and Phylogenetic Analyses

Newly generated sequences were edited using Sequencher 4.1.4 (Gene Codes, Ann Arbor, MI, USA), and haplotypes of heterozygotes were resolved according to Hughes et al. [33]. The missing or ambiguous loci were coded as “N”. Other sequences for phylogenetic analyses were downloaded from GenBank following those used in [22,23,34,35,36,37,38,39,40,41,42,43,44,45,46,47,48] (see Table 1). Nucleotide sequences from each gene were aligned using MAFFT v.7.245 [49], and the ambiguous alignments were adjusted manually with MEGA7 [50]. The alignments of the four loci were uploaded to Zenodo (https://zenodo.org/records/14244210, accessed on 25 November 2024). Given that dataset from different loci, topological incongruence between partitions (nuclear markers vs. mitochondria markers) was examined using the incongruence length difference (ILD) test [51] implemented in PAUP4.0 [52] with 1000 heuristic after the removal of all invariable characters [53]. The maximum likelihood (ML) and Bayesian inference (BI) methods were used to analyze the combined datasets of the four loci with the procedures provided in detail by Zhu & Bau [26]. Four Markov chains were run for 3,000,000 generations with sampling every 100 generations until the split deviation frequency value was less than 0.01 [54]. Bootstrap support (BS) values greater than 70% (RAxML GUI 2.0.0-beta analyses) and Bayesian posterior probabilities (BPP) higher than 0.95 were regarded as significantly supported, and BS between 50% and 70% and BPP between 0.90 and 0.95 were considered as weakly supported; otherwise, the result was listed as unresolved phylogeny [55,56].

**Table 1 jof-10-00835-t001:** Fungal species and sequences used in phylogenetic analyses.

Taxon	Seq.-ID	Location	*ITS*	*nrLSU*	*tef-1α*	*mtSSU*	Reference
*C. aesontiensis*	LZ P-7614 (type)	Italy	KY554753	KY554752			[34]
*C. afronivea*	SFSU BAP 619 (type)	São Tomé	NR_148105				[35]
*C. afronivea*	HMJAU46372	China	MW822049	OL376317			[23]
** *C. afronivea* **	**HMJAU67175**	**China**	**OR921294**				**This study**
** *C. afronivea* **	**HMJAU67192**	**China**	**OR921295**	**OR921344**	**OR940179**	**OR916193**	**This study**
** *C. afronivea* **	**HMJAU67193**	**China**	**OR921297**				**This study**
** *C. afronivea* **	**HMJAU67194**	**China**	**OR921298**				**This study**
** *C. afronivea* **	**HMJAU67195**	**China**	**OR921296**				**This study**
** *C. afronivea* **	**HMJAU67196**	**China**	**OR921282**	**OR921333**	**OR940170**	**OR916192**	**This study**
*C. atramentaria*	SZMC-NL-4245	Hungary	FN396123	FN396347	FN396225		[36]
*C. caribaeonivea*	PA-2023a ANGE1390	Dominican Republic	OQ275140				[37]
*C. aesontiensis*	LZ P-7614 (type)	Italy	KY554753	KY554752			[34]
*C. cerkezii*	CNF1/7253 (type)	Croatia	KX869912	KX869913			[38]
*C. cinerea*	SZMC-NL-2141	Hungary	FN396149	FN396190			[36]
*C. coniophorus*	SZMC-NL-3414	Hungary	FN396122	FN396207			[36]
*C. cortinata*	SZMC-NL-1621	Hungary	FN396121	FN396171	FN396224		[36]
*C. cothurnata*	CBS 174.49	Netherlands	MH856479	MH868018			[39]
** *C. furfuracea* **	**HMJAU67156 (type)**	**China**	**OR921288**	**OR921346**	**OR940182**	**OR916200**	**This study**
** *C. furfuracea* **	**HMJAU67159**	**China**	**OR921290**				**This study**
** *C. furfuracea* **	**HMJAU67160**	**China**	**OR921291**				**This study**
** *C. furfuracea* **	**HMJAU67161**	**China**	**OR921292**				**This study**
** *C. furfuracea* **	**HMJAU67162**	**China**	**OR921293**	**OR921340**	**OR940175**	**OR916199**	**This study**
** *C. furfuracea* **	**HMJAU67163**	**China**	**OR921286**				**This study**
** *C. furfuracea* **	**HMJAU67164**	**China**	**OR921287**				**This study**
** *C. furfuracea* **	**HMJAU67165**	**China**	**OR921289**				**This study**
*C. furfuracea*	RAAA 2021	Iraq	MZ265188				[40]
*C. furfuracea*	Ghobad-Nejhad 4282	Iran	MT535708	MT554301			[40]
*C. igarashii*	CBM-FB38829 (type)	Japan	AB854625				[41]
** *C. igarashii* **	**HMJAU67212**	**China**	**OR921284**	**OR921338**		**OR916188**	**This study**
** *C. igarashii* **	**HMJAU67213**	**Chin**	**OR921326**				**This study**
** *C. igarashii* **	**HMJAU67214**	**China**	**OR921324**				**This study**
** *C. iliensis* **	**HMJAU67171 (type)**	**China**	**OR921305**	**OR921331**	**OR940169**	**OR916195**	**This study**
** *C. iliensis* **	**HMJAU67172**	**China**	**OR921306**	**OR921332**			**This study**
** *C. khorqinensis* **	**HMJAU67147 (type)**	**China**	**OR921330**	**OR921348**	**OR940178**	**OR940168**	**This study**
*C. lagopus*	SZMC-NL-0191	Hungary	JN943127	JQ045867			[42]
*C. marcescibilis*	SZMC-NL-2140	Hungary	FM878020	FM876277	FM897257		[43]
*C. musae*	JV06-179	Sweden	NR_148070	KC992965			[44]
*C. narcotica*	SZMC-NL-2342	Hungary	FM163180	FM160729	FN396290		[45]
*C. nivea*	4585	USA	JF907848				[46]
*C. nivea*	SZMC-NL-0847	Hungary	HQ847032	HQ847117			[36]
** *C. nivea* **	**HMJAU58777**	**China**	**MZ220450**				**This study**
** *C. nivea* **	**HMJAU67201**	**China**	**OR921317**				**This study**
** *C. nivea* **	**HMJAU67145**	**China**	**OR921309**				**This study**
** *C. nivea* **	**HMJAU67202**	**China**	**OR921315**				**This study**
** *C. nivea* **	**HMJAU67203**	**China**	**OR921307**				**This study**
** *C. nivea* **	**HMJAU67204**	**China**	**OR921313**				**This study**
** *C. nivea* **	**HMJAU67205**	**China**	**OR921314**				**This study**
** *C. nivea* **	**HMJAU67206**	**China**	**OR921312**				**This study**
** *C. nivea* **	**HMJAU67207**	**China**	**OR921308**				**This study**
** *C. nivea* **	**HMJAU67208**	**China**	**OR921281**	**OR921339**	**OR940174**	**OR916189**	**This study**
** *C. nivea* **	**HMJAU67209**	**China**	**OR921310**				**This study**
** *C. nivea* **	**HMJAU67210**	**China**	**OR921311**				**This study**
** *C. nivea* **	**HMJAU67211**	**China**	**OR921316**				**This study**
*C. psammophila*	CNF 1/6401	Libya	MK491274	MK492278			[47]
*C. pseudomarcecibilis*	AH:33711	Spain	KY698008	MF033345			[48]
*C. pseudonivea*	SZMC:NL:2340	Hungary	FM163181	FM160728	FN430698		[41]
*C. pseudonivea*	HMJAU46449	China	MW822599	OL376335			[23]
*C. pseudonivea*	HFRG_EJ220922_1_FRDBI 28794927	United Kingdom	OQ133583				Unpublish-ed
** *C. pseudonivea* **	**HMJAU67153**	**China**	**OR921301**	**OR921347**	**OR940183**	**OR916201**	**This study**
** *C. pseudonivea* **	**HMJAU67198**	**China**	**OR921328**				**This study**
** *C. pseudonivea* **	**HMJAU67199**	**China**	**OR921327**				**This study**
** *C. sericivia* **	**HMJAU67200 (type)**	**China**	**OR921285**	**OR921342**	**OR940176**	**OR916190**	**This study**
** *C. sericivia* **	**HMJAU67201**	**China**	**OR921317**				**This study**
*C.* sp. 1	TPN-2017 CBM-FB42007	Vietnam	LC259498				[22]
***C.* sp. 2**	**HMJAU67197**	**China**	**OR921329**	**OR921343**	**OR940177**	**OR916191**	**This study**
***C.* sp. 3**	**HMJAU67173**	**China**	**OR921300**				**This study**
***C.* sp. 3**	**HMJAU67174**	**China**	**OR921299**	**OR921334**	**OR940171**	**OR916194**	**This study**
*C. strossmayeri*	SZMC-NL-0774	Hungary	HQ847048	HQ847129			[34]
** *C. subigarashii* **	**HMJAU67215**	**China**	**OR921322**				**This study**
** *C. subigarashii* **	**HMJAU67216**	**China**	**OR921321**				**This study**
** *C. subigarashii* **	**HMJAU67217**	**China**	**OR921280**	**OR921341**		**OR916187**	**This study**
** *C. subigarashii* **	**HMJAU67218**	**China**	**OR921279**		**OR940180**		**This study**
** *C. subigarashii* **	**HMJAU67219**	**China**	**OR921283**	**OR921345**	**OR940181**	**OR918461**	**This study**
** *C. subigarashii* **	**HMJAU67220**	**China**	**OR921318**				**This study**
** *C. subigarashii* **	**HMJAU67221**	**China**	**OR921319**				**This study**
** *C. subigarashii* **	**HMJAU67222**	**China**	**OR921320**				**This study**
** *C. subigarashii* **	**HMJAU67223 (type)**	**China**	**OR921323**				**This study**
** *C. subigarashii* **	**HMJAU67224**	**China-**	**OR921325**				**This study**
** *C. tenuipes* **	**HMJAU67168**	**China**	**OR921304**	**OR921337**		**OR916198**	**This study**
** *C. tenuipes* **	**HMJAU67169 (type)**	**China**	**OR921303**	**OR921335**	**OR940172**	**OR916197**	**This study**
** *C. tenuipes* **	**HMJAU67170**	**China**	**OR921302**	**OR921336**	**OR940173**	**OR916196**	**This study**
*C. udicola*	AM1240	Germany	KC992967	KC992967	KJ732831		[44]
*C. utrifer*	SZMC-NL-0591	Hungary	FN396140	FN396209			[36]

Newly generated sequences in this study are shown in bold font.

## 3. Results

### 3.1. Molecular Phylogeny

In the concatenated dataset of *ITS* + *LSU* + *tef1-α* + *mtSSU*, a total of 165 sequences (83 for *ITS*, 37 for *LSU*, 20 for *tef1-α*, and 15 for *mtSSU*) from 83 collections were included. The length of aligned dataset was 3980 bp including gaps (704 bp for *ITS*, 1367 bp for *LSU*, 1177 bp for *tef1-α*, and 732 bp for *mtSSU*). *Coprinopsis strossmayeri* (Schulzer) Redhead, Vilgalys & Moncalvo; *C. narcotica*; *C. lagopus* (Fr.) Redhead, Vilgalys & Moncalvo; *C. cinerea* (Schaeff.) Redhead, Vilgalys & Moncalvo; *C. atramentaria*; *C. pammophila*; and *C. cothurnata* were selected as outgroups based on recent studies [22,23]. The ILD test comparing the nuclear markers and mitochondria markers yielded a P value of 0.11, indicating congruence among the nuclear and mitochondria datasets. For this reason, a combined dataset was used for analyses. The best-fit models of the Bayesian analysis of the combined dataset were GTR + F + I + G4 for ITS, LSU, and tef1-α, and HKY + F + I+G4 for mtSSU. The ML analysis resulted a similar topology as the Bayesian topology, and only the former is shown in Figure 1.

The four-loci phylogenetic framework showed that species with globose veil elements were multiphyletic in *Coprinopsis*, and sect. *Niveae* formed a distinct clade in the genus *Coprinopsis* with a high support rate (ML/BI = 100/1.00), containing fourteen phylogenetic species of which six new were well-supported lineages (*Coprinopsis furfuracea*, *Coprinopsis iliensis*, *Coprinopsis tenuipes*, *Coprinopsis subigarashii*, *Coprinopsis sericivia*, and *Coprinopsis khorqinensis*). Among these species, *C. furfuracea* was at the base status of this section and sister to the clade composed of the other species. *C. iliensis* and *C. tenuipes* composed a clade and were sister to each other. *Coprinopsis afronivea*, *C. caribaeonivea*, and *C.* sp. 2 were clustered into a lignicolous clade, which differed from other fimicolus species. In addition to these species, other species of this section, including *C. nivea*, shared a close genetic relationship, which corresponds to their easily confused macroscopic morphology.

The phylogenetic tree based on analysis of *ITS* or combined *ITS*-*LSU* date showed similar topology with the four-loci dataset. It is noteworthy that the voucher HMJAU67170 clustered into *C. pseudonivea* based on the single-locus or di-loci dataset, but with *C. tenuipes* in the four-loci dataset. Combined with the morphological results, the features of this voucher were consistent with *C. tenuipes*.

### 3.2. Taxonomy

***Coprinopsis* sect. *Niveae*** (Citérin) D.J. Schaf.

**Description:** Basidiomata small to medium-sized, mostly fimicolous, rare lignicolous or terrestrial. Pileus campanulate, broad conical or semiglobose, or plano-convex when mature, but never applanate, white, light grey to grey, or light pinkish brown, completely covered by dense, white or cream, powdery, or furfuraceous veil at center and hairy–floccose veil at margin, occasionally with radiate plication. Lamellae free, but absent ring-zone, deliquescent or withering. Stipes white, without volva-like margin at stipe, covered by sparse, white floccose veil. Basidiospores medium- to large-sized, usually flattened, subglobose to ellipsoid, or rounded hexagon with apical papilla in frontal view, ellipsoid to oblong inside view, germ pore central or slightly eccentric. Basidia dimorphological or trimorphological, 4-spored. Cheilocystidia subglobose, utriform, or broad fusiform. Pleurocystidia present or absent, when present, broad lageniform or subcylindrical. Veil elements mostly globose to subglobose, thin-walled to slightly thick-walled, colorless to cream, sometimes encrusted, chains of subcylindrical elements also present, sometimes branched. Clamp connections present.

**Note.** Though the establishment of sect. *Niveae* and sect. *Subniveae* has been confirmed by several studies, the clear distinctions between these two sections have not yet been proposed. In sect. *Niveae*, when mature, the pileus is subglobose or campanulate to plano-convex and sometimes with subumbonate apex, but never applanate. However, the pileus of species in sect. *Subniveae* is applanate and with a depressed central part at age. Moreover, the veil on the pileus is dense in species in sect. *Niveae*, while this structure in sect. *Subniveae* is relatively sparse [16,38,57].

***Coprinopsis furfuracea*** T. Bau & L.Y. Zhu sp. nov Figure 2E–H, Figure 3e1,e2 and Figure 4

**Mycobank:** MB855021.

**Etymology:** “*furfuracea*” refers to its furfuraceous veil residues on the pileus.

**Diagnosis:** Basidiomata medium-sized; pileus light grey, almost applanate when mature, with radiate plication up to 1/4 part to the top of pileus, covered with white to light brown furfuraceous veil residues; lamellae free but without ring-zone, deliquescent or withering at aged; basidiospores ellipsoid, 11.1–15.6 × 8.2–8.6 × 7.0–8.7 μm, germ pore eccentric; cheilocystidia subglobose, utriform, or broad fusiform; pleurocystidia absent

**Type:** China: Inner Mongolia: Hulunbuir City, Chen Barag Banner, State-run Hadatu grazing land, in small groups on sheep dung, 9 August 2022, Tolgor Bau and Li-Yang Zhu, HMJAU67156.

**Description:** Basidiomata small to medium-sized. Pileus 0.7–1.2 × 0.8–1.0 cm when mature, first ovoid or ellipsoid, then obtusely conical, finally plano-convex, sometimes margin involved; first cream to light brown when young, then white to greyish white to light grey; covered with cream to light brown furfuraceous veil at central; radiate plication present, up to 1/4 part to the top of pileus. Lamellae free, but absent ring-zone, L = 39–46, I = 0–3, white first, dark grey-black when mature, deliquescent or withering. Context subcarnulosus. Stipes 4.2–6.2 × 0.3–0.4 cm, equal or tapering upwards, white to cream, covered by white to cream floccose veil. Smell somewhat like rotten grass, odor unknown.

Basidiospores [60, 4, 4] (11.1)13.0–13.4(15.6) × (7.5)8.2–8.6(10.1) × (7.0)7.6–7.9(8.7) μm, in average 13.2 × 8.4 × 7.8 μm, Q1 = 1.30–1.76, Q2 = 1.58–1.93, ellipsoid to oblong without apical papilla in frontal and side view, brown in H_2_O, almost dark grey-brown in 5% KOH solution; germ pore eccentric, 1.4–2.7 μm in width. Basidia trimorphological, clavate to long clavate, usually contracted in the middle part, 24–46 × 9–12 μm (elongated basidia 38–46 × 9–12 μm, medium basidia 29–34 × 9–12 μm, and short basidia 24–27 × 10–11 μm), 4-spored, sterigmata 3–8 μm, surrounded by 4–6 pseudoparaphyses; hyphae of lamellae trama regular, colorless, 4–7 μm in width. Cheilocystidia subglobose, utriform, or broad fusiform, 30–52 × 16–41 μm, colorless, thin-walled. Pleurocystidia absent. Pileipellis a cutis composed of (sub)cylindrical cells, 38–124 × 6–20 μm. Hyphae of caulopellis and caulotrama 8–15 μm and 17–30 μm in width, respectively. Veils on top of pileus mostly composed of (sub)globose cells, 26–87 × 21–85 μm, colorless or brown-yellow, present tiny instruction or granular, thin-walled to slightly thick-walled; veils at margin of pileus and on stipes composed of chains of narrow (sub)cylindrical elements, sometimes branched, 3–9 μm in width, colorless or brown-yellow, thin-walled to slightly thick-walled. Clamp connection present but rare.

**Habitat:** Subfasciculate or in small groups on sheep dung or enriched soil. Usually occurs in summer and early autumn.

**Distribution:** Temperate to subtropical area of Asia.

**Other specimens examined.** Inner Mongolia: Same location with type specimens, 9 August 2022, Tolgor Bau, Li-Yang Zhu, HMJAU67157, HMJAU67158; Wei-Nan Hou HMJAU67159, HMJAU67160, Han-Bing Song, HMJAU67161, HMJAU67162, HMJAU67166; Shi-En Wang, HMJAU67163, HMJAU67164, HMJAU67167; Hubei: Shennongjia Forest District, Muyu Town, in clustered on nutrient-rich soil, Li-Yang Zhu, 24 June 2022, HMJAU67165.

**Note.** *C. furfuracea* was previously invalidly published by Al Anbagi et al. [38], named as *C. iraqicus*, as this name was not registered in any of the three recognized repositories (Fungal Names, Index Fungorum, or MycoBank) of International Code of Nomenclature for Algae, Fungi, and Plants [ICN] Art. F.5.2 [58,59]. As there is no technical or nomenclatural reason, the invalidity of “*C. iraqicus*” requires a new, valid taxon name rather than a new replacement name, accompanied with a new, separate identifier [Art. 6.9, Art. F. 5.1] [59,60]. As the former epithet, “*iraqicus*”, was named after the type location, which could not match with the wider distribution and characteristics of this species, we chose “*furfuracea*” as the new epithet for the distinguishing features of this species.

*Coprinopsis furfuracea* is characterized by its radiate grooves of the pileus, cream to light brown furfuraceous veils, and ellipsoid basidiospores. Uljé & Nooderloos [16] illustrated *C. nivea* as also with a pileus with plication, while from our observation, this species lacks these characteristics; in consideration, the length range of basidiospores (12.2–19.0 μm) might contain at least two species, this description might not refer to the actual *C. nivea*.

*Coprinopsis furfuracea* is the basal species of sect. *Niveae*, which retains similar characteristics to the species of the sister group, sect. *Subniveae*, such as radiate plication on the pileus, almost applanate pileus when mature, ellipsoid basidiospores, and veil elements with brown hue [22]. However, the pileus of this species is rather larger and without a sunken disc-like central part and they are mostly coprophilous, which resembles species to species in sect. *Niveae*.

***Coprinopsis iliensis*** T. Bau & L.Y. Zhu sp. nov Figure 2O,Q, Figure 3j1,j2 and Figure 5

**Mycobank:** MB855022.

**Etymology:** “*iliensis*” refers to its known location, Ili Valley in Xinjiang Uygur Autonomous Region of China.

**Diagnosis:** Basidiomata middle to large-sized; pileus white, submembrane, covered with white powdery veil residues; lamellae withering at aged; basidiospores ellipsoid or subcampanulate in front view, 14.0–18.7 × 10.4–12.4 × 8.7–10.8 μm, germ pore central; cheilocystidia subglobose, utriform, broad ellipsoid, or subfusiform; pleurocystidia absent.

**Type:** China: Xinjiang: Yining City, Turkes County, Aketas Girl Peak, grows in alpine grassland, 13 August 2023, Hong Cheng and Hanbing Song, HMJAU67171.

**Description:** Basidiomata medium or large-sized. Pileus 2.6–3.4 × 1.5–1.9 cm when mature, first ovoid or ellipsoid, obtusely conical when mature, never applanate and sometimes margin involved; submembrane, white when young, white to grey-white at age, without any brown hue on pileus; covered with white powdery veil; radiate plication absent. Lamellae adnexed, L = 48–62, I = 0–2, white at first, dark greyish brown to almost black when mature, withering. Context in central subcarnulosus and at margin very thin. Stipes 9.2–13.5 × 0.4–0.5 cm, equal or tapering upwards, white, covered by sparse white floccose veil. Smell not obvious, odor unknown.

Basidiospores [60, 2, 2] (14.0)16.5–16.9(18.7) × (10.4)11.1–11.4(12.4) × (8.7)9.6–9.9(10.8) μm, in average 16.7 × 11.2 × 9.7 μm, Q1 = 1.29–1.63, Q2 = 1.67–1.87, ellipsoid to oblong in frontal view, oblong in side view, dark red-brown in H_2_O, almost black in 5% KOH solution; germ pore central, 2.2–3.7 μm in width. Basidia trimorphological, short clavate to clavate, sometimes constricted in middle part, 21–46 × 11–15 μm (elongated basidia 38–46 × 12–15 μm, medium basidia 28–35 × 12–14 μm, and short basidia 21–25 × 12–15 μm), 4-spored, sterigmata 4–7 μm, surrounded by 4–6 pseudoparaphyses; hyphae of lamellae trama regular, colorless, 2–9 μm in width. Cheilocystidia present but rare, subglobose, utriform, or subfusiform, 43–68 × 24–51 μm, colorless, thin-walled. Pleurocystidia absent. Pileipellis a cutis, composed of (sub)cylindrical cells, 33–104 × 9–18 μm. Hyphae of caulopellis and caulotrama 4–8 μm and 12–23 μm in width, respectively. Veils on top of pileus mostly composed of (sub)globose cells, 13–40 × 13–38 μm, smooth, thin-walled to slightly thick-walled; veils at margin of pileus and on stipes composed of chains of narrow (sub)cylindrical elements, sometimes branched, 4–8 μm in width. Clamp connection present.

**Habitat:** Single or spreading grows in alpine grassland. Occur in summer.

**Distribution:** only known from type location.

**Other specimens examed:** same location with type specimens, Hong Cheng and Han-Bing Song, 13 August 2023, HMJAU67172.

**Notes.** This species is similar to *C. nivea* in macromorphology, but the latter has thicker context, rounded hexagonal or limoniform basidiospores, and subcylindrical pleurocystidia [22]. *C. tenuipes* is sister to this specie, while the light greyish brown pileus and the smaller size of basidiospores (up to 12 μm in length) could easily distinguish the former from *C. iliensis*.

***Coprinopsis tenuipes*** T. Bau & L.Y. Zhu sp. nov Figure 2P,R, Figure 3h1,h2 and Figure 6

**Mycobank:** MB855023.

**Etymology:** “*tenuipes*” refers to its slender stipes.

**Diagnosis:** Basidiomata medium-sized; pileus light-grey, thin, with fibrous veil residues at margin of pileus; stipes slender; basidiospores subglobose in front view and ellipsoid in side view, 9.5–12.3 × 8.7–10.7 × 6.3–8.3 μm, germ pore central; cheilocystidia present but rare, subglobose, utriform, or broad ellipsoid; pleurocystidia absent.

**Type:** China: Yunnan: Chuxiong Yi Autonomous Prefecture, Mouding County, Gonghe Town, Zhuoguan Mountain, in small groups on cow dung, 24 July 2023, Kai-Lin Liu, HMJAU67169.

**Description:** Basidiomata medium-sized. Pileus 1.6–2.5 × 0.6–1.0 cm when mature, first ovoid, ellipsoid, then obtusely conical to convex, finally plano-convex with reflexed margin, subumbonate at center; white to greyish white when young, greyish white to light grey at age, sometimes with light brown hue at top of pileus; covered with dense, white to cream powdery veil at central and fibrous veil at margin; radiate plication absent. Lamellae adnexed, L = 35–39, I = 1–3, white first, then grey to black, withering or deliquescent at margin. Context white to light grey, very thin. Stipes 7.5–11.2 × 0.2–0.3 cm, equal or attenuate upwards, white to cream, covered by white to light grey floccose veil. Smell and odor unknown.

Basidiospores [60, 4, 2] (9.5)11.0–11.3(12.3) × (8.7)9.5–9.8(10.7) × (6.3)7.2–7.5(8.3) μm, in average 11.2 × 9.7 × 7.3 μm, Q1 = 1.03–1.35, Q2 = 1.38–1.69, flattened, subglobose, or rounded hexagon with apical papilla in frontal view, ellipsoid in side view, dark red-brown in H_2_O, almost black in 5% KOH solution; germ pore central, 1.9–3.3 μm in width. Basidia trimorphological, short clavate to clavate, sometimes constricted in middle part, 21–39 × 8–10 μm (elongated basidia 37–39 × 9–10 μm, medium basidia 29–34 × 8–10 μm, and short basidia 21–25 × 8–10 μm), sterigmata 4–6 μm, 4-spored, surrounded by 3–6 pseudoparaphyses; hyphae of lamellae trama regular, colorless, 5–14 μm in width, sometimes diverticulate. Cheilocystidia present but rare, utriform, subglobose to broad ellipsoid, 25–51 × 25–29 μm, colorless, thin-walled. Pleurocystidia absent. Pileipellis a cutis, composed of (sub)cylindrical cells, 9–15 μm in width, sometimes diverticulate, subpileipellis not obvious. Hyphae of caulopellis and caulotrama 9–15 μm and 15–32 μm in width, respectively. Veils on top of pileus mostly composed of (sub)globose cells, 15–40 × 14–39 μm, almost smooth, obvious instruction unseen, thin-walled to slightly thick-walled; veils at margin of pileus and on stipes composed of chains of narrow (sub)cylindrical elements, sometimes branched, 6–11 μm in width. Clamp connection present but rare, mostly found in stipipellis.

**Habitat:** in small groups grows on cow dung. Usually occurs in summer.

**Distribution:** So far known from Southwest China.

**Other specimens examined:** Guangxi: Hechi City, Huanjiang Maonan Autonomous County, Mulun National Nature Reserve, in small groups on cow dung, Guang-Fu Mou, 9 April 2021, HMJAU67168 (M2021040927).

**Notes:** Macromorphologically, *C. tenuipes* resembles *C. igarashii*, and these two species both grow on cow dung; however, the latter have utriform, cylindrical or broad ellipsoid pleurocystidia; additioanlly, the latter is distributed in the higher latitudes of the North Temperate Zone in Asia (Hokkaido in Japan and Northeast and Northwest China), as now known [41]. *C. iliensis* is sister to this species; they both have a thin pileus, subglobose or utriform cheilocystidia, and lack pleurocystidia, while the pileus of *C*. *iliensis* is almost white even when mature, and its basidiospores are ellipsoid or limoniform in front view and relatively larger with size 14.0–18.7 × 10.4–12.4 × 8.7–10.8 μm. There might be horizontal gene transfer in ITS between *C. tenuipes* and *C. pseudonivea*; however, *C. pseudonivea* processes a light pink-brown pileus and subcylindrical pleurocystidia [16,23].

***Coprinopsis sericivia*** T. Bau & L.Y. Zhu sp. nov Figure 2I–K, Figure 3g1,g2 and Figure 7

**Mycobank:** MB855024.

**Etymology:** “*sericivia*” refers to the species distributed along the ancient Silk Road in China.

**Diagnosis:** Basidiomata small to medium-sized; pileus white, semiglobose to obtusely conical at aged, covered with white powdery veil; lamellae free, deliquescent when old; basidiospores subglobose, rhomboid, or limoniform in front view and ellipsoid to oblong in side view, 14.6–18.8 × 11.2–13.8 × 8.2–10.6 μm, germ pore slightly eccentric; cheilocystidia utriform, broad ellipsoid, broad lageniform, or subcylindrical; pleurocystidia broad lageniform or subcylindrical, less than 100 μm in length.

**Type:** China: Xinjiang: Altay Region, Buerjin County, Hemu Kanas Mongolian Town, Hemu County, in small groups on horse dung, 17 August 2021, Qian-Qi Ye and Xiao-Liang Liu, HMJAU67200.

**Description:** Basidiomata small to medium-sized. Pileus 1.3–2.4 × 0.6–0.8 cm when mature, first ovoid or ellipsoid, semiglobose to obtusely conical at aged, never applanate; white from young to mature; covered with white powdery veil at central and fibrous veil at margin; sometimes with radiate grooves at margin of pileus. Lamellae free, L = 27–41, I = 3, white first, then dark greyish black to almost black, deliquescent. Context thin. Stipes 4.7–6.9 × 0.3–0.4 cm, equal, white, covered by white floccose veil. Smell not obvious, odor unknown.

Basidiospores [60, 4, 2] (12.5)14.6–18.8(19.2) × (10.0)11.2–13.8(4.0) × (7.8)8.2–10.6(11.8) μm, in average 16.7 × 12.6 × 9.6 μm, Q1 = 1.14–1.47, Q2 = 1.49–1.88, flattened, subglobose, rhomboid or limoniform with apical papilla in frontal view, ellipsoid to oblong in side view, dark red-brown in H_2_O, almost black in 5% KOH solution, smooth; germ pore slightly eccentric, 1.9–3.7 μm in width. Basidia trimorphological, short clavate to clavate, usually constricted in middle part, 16–35 × 11–14 μm (elongated basidia 31–35 × 11–13 μm, medium basidia 23–26 × 11–14 μm, and short basidia 16–21 × 11–13 μm), 4-spored, sterigmata 5–7 μm, surrounded by 4–6 pseudoparaphyses; hyphae of lamellae trama regular, colorless, 3–5 μm in width. Cheilocystidia broad utriform, broad ellipsoid, broad lageniform, 23–83 × 12–40 μm, colorless, thin-walled. Pleurocystidia broad lageniform or subcylindrical, 48–91 × 18–40 μm, colorless, thin-walled. Pileipellis a cutis, composed of (sub)cylindrical cells, 5–26 μm in width, subpileipellis present. Hyphae of caulopellis and caulotrama 5–12 μm and 19–35 μm in width, respectively. Veils on top of pileus mostly composed of (sub)globose to ellipsoid cells, 18–42 × 13–35 μm, instructed, thin-walled to slightly thick-walled, colorless to light greyish brown; veils at margin of pileus and on stipes composed of chains of narrow (sub)cylindrical elements, multibranched, 2–8 μm in width. Clamp connection present but rare.

**Habitat:** In small groups on horse dung. Usually occurs in summer and early autumn.

**Distribution:** Only known from the northwestern part of Xinjiang Uygur Autonomous Region of China.

**Other specimens examined:** Xinjiang: Yining City, Turkes County, Aketas Girl Peak, grows in alpine grassland, 13 August 2023, Hong Cheng and Han-Bing Song, HMJAU67172; Bortala Mongol Autonomous Prefecture, Bole City, Sayram Lake, on horse dung, 20 June 2024, Qing-Qing Dong and Qian-Qi Ye, HMJAU67543.

**Notes:** *C. sericivia* is morphologically similar to *C. nivea*, *C. iliensis*, and *C. furfuracea* for their snow-white pileus. Differing from *C. sericivia*, *C. iliensis* lacks pleurocystidia. *C. furfuracea* grows on sheep dung and its pileus covered with cream to light brown furfuraceous veils. *C. nivea* mostly grows on cow dung, and the pileus of *C. nivea* is most cylindrical–ellipsoid, and its pleurocystidia are larger, with a length up to 150 μm [16].

***Coprinopsis subigarashii*** T. Bau & L.Y. Zhu sp. nov Figure 2V, Figure 3i1,i2 and Figure 8

**Mycobank:** MB855026.

**Etymology:** “*subigarashii*” means this species sharing a close relationship with *C. igarashii*.

**Diagnosis:** Basidiomata small to medium-sized; pileus white, semiglobose to obtusely conical when aged, covered with white powdery veil; lamellae free, deliquescent when old; basidiospores subglobose, rhomboid, or limoniform in front view and ellipsoid to oblong in side view, 10.9–12.2 × 8.4–10.5 × 6.4–7.3 μm, germ pore slightly eccentric; basidia dimorphological; cheilocystidia utriform, broad ellipsoid, broad lageniform, or subcylindrical; pleurocystidia broad lageniform or subcylindrical, less than 100 μm in length.

**Type:** China: Inner Mongolia: Xing’an League, Tuquan County, East Durkheim State Farm Eleven Team, solitary or in small groups on sheep dung, 26 August 2023, Li-Yang Zhu and Wei-Nan Hou, HMJAU67223.

**Description:** Basidiomata small to medium-sized. Pileus 1.3–2.4 × 0.6–0.8 cm when mature, first ovoid or ellipsoid, semiglobose to obtusely conical at aged, never applanate; white from young to mature; covered with white powdery veil at central and fibrous veil at margin; sometimes with radiate grooves at margin of pileus. Lamellae free, L = 27–41, I = 3, white first, then dark greyish black to almost black, deliquescent. Context thin. Stipes 4.7–6.9 × 0.3–0.4 cm, equal, white, covered by white floccose veil. Smell not obvious, odor unknown.

Basidiospores [80, 5, 4] (9.9) 10.9–12.2 (12.8) × (8.0)8.4–10.5(11.1) × (6.0)6.4–7.3(7.7) μm, in average 11.5 × 9.6 × 6.8 μm, Q1 = 1.05–1.44, Q2 = 1.49–1.96, flattened, subglobose, rhomboid, or limoniform with apical papilla in frontal view, ellipsoid to oblong in side view, dark red-brown in H_2_O, almost black in 5% KOH solution, smooth; germ pore slightly eccentric, 1.4–2.8 μm in width. Basidia dimorphological, short clavate to clavate, usually constricted in middle part, 14–30 × 9–12 μm (elongated basidia 24–30 × 9–11 μm and short basidia 14–21 × 9–12 μm), 4-spored, sterigmata 3–5 μm, surrounded by 4–6 pseudoparaphyses; hyphae of lamellae trama regular, colorless, 2–7 μm in width. Cheilocystidia subglobose, broad utriform, broad ellipsoid, broad lageniform, 22–60 × 13–36 μm, colorless, thin-walled. Pleurocystidia broad lageniform or subcylindrical, 49–109 × 23–45 μm, colorless, thin-walled. Pileipellis a cutis, composed of (sub)cylindrical cells, 5–15 μm in width, subpileipellis present. Hyphae of caulopellis and caulotrama 4–13 μm and 15–38 μm in width, respectively. Veils on top of pileus mostly composed of (sub)globose to ellipsoid cells, 14–46 × 14–36 μm, instructed, thin-walled to slightly thick-walled, colorless to light greyish brown; veils at margin of pileus and on stipes composed of chains of narrow (sub)cylindrical elements, multibranched, 2–10 μm in width. Clamp connection present.

**Habitat:** Usually on cow dung, occasionally wild boar dung. Usually occurs in summer and early autumn.

**Distribution:** So far known from northern part of China.

**Other specimens examined:** Jilin: Baicheng City, Tongyu County, Xianghai National Reserve, on cow dung, Tolgor Bau and Li-Yang Zhu, 19 August 2021, HMJAU67220, HMJAU67221; Yanbian Korean Autonomous Prefecture, Longjing City, Baijin Forest Farm, on wild boar (*Sus scrofa*) dung, Tolgor Bau and Ya-Li Sun, 24 August 2020, HMJAU67217; Inner Mongolia: Tongliao City, Khorqin Left Back Banner, Jinbaotun Town, on cow dung, Tolgor Bau and Li-Yang Zhu, 7 August 2021, HMJAU67218, HMJAU67219; Hulunbuir City, Chen Barag Banner, on cow dung, 9 August 2022, Tolgor Bau and Shi-En Wang, HMJAU67216; same location, on cow dung, 9 August 2022, Tolgor Bau and Li-Yang Zhu, HMJAU67222; Xinjiang: Yining City, Turkes County, Yelang Velley, on cow dung, 11 August 2023, Hong Cheng, Han-Bing Song and Xian-Yan Zhou, HMJAU67215, HMJAU67224.

**Notes:** Sister to *C. subigarashii*, *C. igarashii* has trimorphological basidia and basidiospores, which are five-angular or heart-shaped in frontal view and with a rough surface composed of tiny warts [41]. *C. subigarashii* is also morphologically similar to *C. nivea*, *C. iliensis*, and *C. furfuracea* for their snow-white pileus. Differing from *C. subigarashii*, *C. iliensis* lacks pleurocystidia. *C. furfuracea* grows on sheep dung and its pileus covered with cream to light brown furfuraceous veils. *C. nivea* mostly grows on cow dung, and the pileus of *C. nivea* is most cylindrical–ellipsoid, and its pleurocystidia are larger, with a length up to 150 μm [16].

***Coprinopsis khor**qinensis*** T. Bau & L.Y. Zhu sp. nov Figure 2T, Figure 3b1,b2 and Figure 9

**Mycobank:** MB855025.

**Etymology:** “*khorqinensis*” refers to its known location, Khorqin sandy land.

**Diagnosis:** Basidiomata small or medium-sized; pileus light-grey, never applanate when mature, covered with powdery or furfuraceous veil residues; lamellae deliquescent when aged; basidiospores subglobose or limoniform in front view and ellipsoid to oblong in side view, 14.1–17.0 × 11.2–13.1 × 8.8–11.3 μm, germ pore slightly eccentric; cheilocystidia utriform, broad ellipsoid, broad lageniform, or subcylindrical; pleurocystidia broad ellipsoid or subcylindrical.

**Type:** China: Jilin: Baicheng City, Tongyu Courty, Xianghai National Nature Reserve, in small groups on mixer of crow dung and straw, 25 August 2023, Li-Yang Zhu and Junlin Wei, HMJAU67147.

**Description:** Basidiomata small or medium-sized. Pileus 1.1–2.4 × 1.6–1.9 cm when mature, first ovoid or ellipsoid, then campanulate, finally obtusely conical, never applanate; white to greyish white when young, greyish white to light grey at age, sometimes with light brown hue at top of pileus; covered with white to cream powdery powdery or furfuraceous veil at central and fibrous veil at margin; radiate plication absent. Lamellae adnexed, L = 34–41, I = 1–3, white first, then dark purple-brown to almost black, deliquescent. Context subcarnulosus. Stipes 4.3–6.2 × 0.3–0.4 cm, equal, white to cream, covered by white to light grey floccose veil. Smell not obvious, odor unknown.

Basidiospores [40, 3, 2] (14.0)15.6–16.0(17.0) × (11.2)12.3–12.7(13.1) × (8.8)9.5–10.2(11.3) μm, in average 15.8 × 12.6 × 9.9 μm, Q1 = 1.14–1.40, Q2 = 1.47–1.89, flattened, subglobose or limoniform with apical papilla in frontal view, ellipsoid to oblong in side view, dark red-brown in H_2_O, almost black in 5% KOH solution; germ pore slightly eccentric, 1.8–3.9 μm in width. Basidia trimorphological, short clavate to clavate, 18–41 × 11–15 μm (elongated basidia 36–41 × 12–15 μm, medium basidia 26–31 × 11–13 μm, and short basidia 18–23 × 11–14 μm), 4-spored, sterigmata 4–8 μm, surrounded by 4–6 pseudoparaphyses; hyphae of lamellae trama regular, colorless, 2–9 μm in width. Cheilocystidia utriform, broad ellipsoid, broad lageniform or subcylindrical, 25–66 × 13–23 μm, colorless, thin-walled. Pleurocystidia broad ellipsoid or subcylindrical, 64–117 × 22–38 μm, colorless, thin-walled. Pileipellis a cutis, composed of (sub)cylindrical cells, 4–13 μm in width, subpileipellis not obvious. Hyphae of caulopellis and caulotrama 6–11 μm and 16–28 μm in width, respectively. Veils on top of pileus mostly composed of (sub)globose cells, 15–78 × 13–58 μm, almost smooth, present tiny instruction or granular, thin-walled to slightly thick-walled; veils at margin of pileus and on stipes composed of chains of narrow (sub)cylindrical elements, sometimes branched, 3–9 μm in width. Clamp connection present but rare.

**Habitat:** single or in small groups cow dung or mixer of cow dung and straw. Usually occurs in summer and early autumn.

**Distribution:** only known from Khorqin sandy land in Northeast China.

**Notes.** *C. khorqinensis* is closely related to *C. igarashii* and *C. pseudonivea*; they both have a white to pale grey pileus and (sub)cylindrical pleurocystidia; however, *C. igarashii* has small-sized basidiocarps, and pileus of *C. pseudonivea* is with an obvious pink-brown hue. Additionally, basidiospores of *C. khorqinensis* are much larger (in average 15.8 μm) than spores of the other two species (up to 12.8 μm for *C. igarashii* and 13.5 μm for *C. pseudonivea*) [16,23,41,61,62].

### 3.3. Key to Species of Coprinopsis sect. Niveae



**1**
Growing on little dead branches of deciduous trees or other plant residues ………………………………………… 2
**1′**
Growing on herbivore dung or nutrient-rich soil …………………………………………………………………………3
**2**
Diameter of pileus veil cells less than 40 μm ………………………………………………………………… *C. afronivea*
**2′**
Diameter of pileus veil cells more than 40 μm ……………………………………………………………*C. caribaeonivea*
**3**
Basidiomata small-sized, diameter of pileus usually less than 1.0 cm ………………………………………………… 4
**3′**
Basidiomata middle- to large-sized, diameter of pileus usually larger than 1.5 cm. …………………………………6
**4**
Central of pileus with light brown hue or pink-brown hue. ………………………………………………*C. pseudonivea*
**4′**
Central of pileus (almost) without light brown hue or pink-brown hue. ………………………………………………5
**5**
Basidiopores five-angular or heart-shaped in frontal view, which with rough surface.………………… *C. igarashii*
**5′**
Basidiopores subglobose, rhomboid or limoniform in frontal view, which with smooth surface. ……*C. subigarashii*
**6**
Pileus with radiate plication. ……………………………………………………………………………………*C. furfuracea*
**6′**
Pileus without radiate plication. ……………………………………………………………………………………………7
**7**
Pileus snowy white when mature. …………………………………………………………………………………………8
**7′**
Pileus greyish white when mature. ………………………………………………………………………………………10
**8**
Growing on horse dung. …………………………………………………………………………………………*C. sericivia*
**8′**
Growing on cow dung. ………………………………………………………………………………………………………9
**9**
Pleurocystidia present. ………………………………………………………………………………………………*C. nivea*
**9′**
Pleurocystidia absent. ………………………………………………………………………………………………*C. iliensis*
**10**
Length of basidiospores less than 10 μm.……………………………………………………………………… *C. tenuipes*
**10′**
Length of basidiospores 11–14 μm. ……………………………………………………………………………*C. khorqiensis*


## 4. Discussion

### 4.1. Species Delimitation of Coprinopsis sect. Niveae

We were originally perplexed by the attribution of specimen HMJAU67170. Using the ITS fragment only, this specimen was grouped with *Coprinopsis pseudonivea*; however, this specimen formed a cluster within *C. tenuipes* upon analysis with the combined *ITS*-*nrLSU*-*tef-1α*-*mtSSU* dataset. As there were no pleurocystidia found in this specimen, which aligned with *C. tenuipes*, we thought the result of the four-gene combined dataset was more reliable.

Our results suggest that the delimitation of the species in sect. *Niveae* requires a combination of morphological, ecological, and multigenic phylogenetic features. Among macro-morphological characteristics, basidiomata size, pileus color, and the presence or absence of plication at the pileus could be mainly used for species identification. Microscopically, the main characteristics are the shape and size of basidiospores and the presence or absence of pleurocystidia. Notably, some species in sect. *Niveae* show an obvious substrate preference, so attention should be given to recording in the field the type of animal associated with the feces.

### 4.2. Revision of Classification of Fecal Species in sect. Niveae Based on Ecological and Geographical Features

Within sect. *Niveae*, *Coprinopsis nivea*, *C. pseudonivea*, and *C. afronivea* are likely to have a wide distribution. Taking *C. nivea* as an example, we combined data from iNaturalist (www.iNaturalist.org) and GBIF (www.gbif.org), and found that it is a globally distributed species, but mainly distributed in Europe and Asia. Additionally, there is evidence suggesting that the photographs of this species are mainly taken in cattle breeding areas but not on the feces of native herbaceous animals in other continents. The undomesticated extinct wild cousins of modern domestic cattle, *Bos taurus primigenius*, originated in Central Asia and the domestication of *Bos taurus* originated in the Near East and North Africa, and then spread to the entire Eurasian continent, and finally reached America with the European settlement [63,64,65]. Therefore, we speculate that the actual origin of *C. nivea* is still Eurasia, and the wider distribution of this species around the world is accompanied by the expansion of human cattle husbandry. There is also a similar phenomenon with *C. pseudonivea*. On the other hand, *C. furfuracea* is found in East Asia and Central Asia, whereas *C. iliensis*, *C. igarashii*, *C. khorqinensis*, *C. sericivia*, *C. tenuipes*, and *C. subigarashii* are exclusively found in East Asia. Among these species with limited distribution, *C. igarashii* and *C. subigarashii* do not show any specific preference for substrates. Other species, however, tend to choose specialized herbivore feces, especially in locations with multispecies pastoral farming practices.

Unfortunately, there has not been any research focusing on the correlation between fecal types and coprophilous fungal species, especially those within Basidiomycota. Nevertheless, some related studies have revealed some indications. Angel and Wicklow found that fungi populations on ruminants (Bovidae) were similar in species composition in the grassland, while those on pronghorn and small-mammal feces showed rather minor similarity [66]. Kruys and Ericson thought that some fecal ascomycetes species were more associated with habitat and food choice of herbivores rather than the dung type/animal species; however, they also found in their studies that more than half of the species occurred on only one substrate [67]. Lokare and Fatima discovered that some coprophilous coprinoid fungi may have preference for certain dung types [68]. Our field investigation in the pastoral area also found that in the same pasture with multiple herbivores, some species showed obvious substrate preferences. For example, *C. nivea*, *C. pseudonivea*, *C. tenuipes*, *C. khorqinensis*, and *C. iliensis* were only found on cow dung, *C. furfuracea* was only found on sheep dung, and *C. sericivia* was only found on horse dung. Taking *C. sericivia* and *C. nivea* as an example of sister taxa, both are often found in the same meadow, so geographic isolation is not the main driver in the divergence between these two species. Additionally, since both cows and horses are herbivores that predominantly graze on low-growing herbaceous plants, the differing substrate properties arising from their dietary preferences cannot be regarded as the most important factor for their divergence. Therefore, we need to focus on the animals, particularly their digestive systems, because the germination of coprophilous fungal spores requires pre-treatment by the mammalian digestive systems, which are also harsh conditions for basidiospore survival and germination [69,70]. Cows are classified within the order Artiodactyla and the family Bovidae, characterized as ruminants possessing four stomachs [71]; in contrast, horses belong to the order Perissodactyla and the family Equidae, lacking rumination and possessing a single stomach complemented by a well-developed cecum [72]. The gastrointestinal environments of these two animals are markedly distinct. We hypothesize that the prolonged selection of distinct digestive tract settings encountered by the common ancestor of *C. sericivia* and *C. nivea* resulted in effective isolation, culminating in speciation. *C. tenuipes*, *C. ilienesis*, and *C. furfuraceae* demonstrate alterations in substrate; nevertheless, geographical isolation factors must also be taken into account. It should be noted that the cattle raised in current livestock farming have undergone complex intra- and interspecies hybridization, and each breed of cattle has its own unique evolutionary and domestication history [73,74,75]. In addition, whether herbivores co-evolve with fecal fungi remains to be further studied.

Lundqvist [76] categorized fecal fungi based on their geographical distribution and substrates into three groups: (1) those with a wide ecological range and low preference for particular substrate; (2) those with a wide ecological range but with a high preference for a particular substrate, and (3) fastidious species restricted to particular substrates. Our study indicates that the third classification of the abovementioned fecal fungi proposed by Lunqvist should be revised to two types, which are (3) those with a narrow ecological range and with a high substrate preference, and (4) those with a narrow ecological range and with low preference for a particular substrate.

## Figures and Tables

**Figure 1 jof-10-00835-f001:**
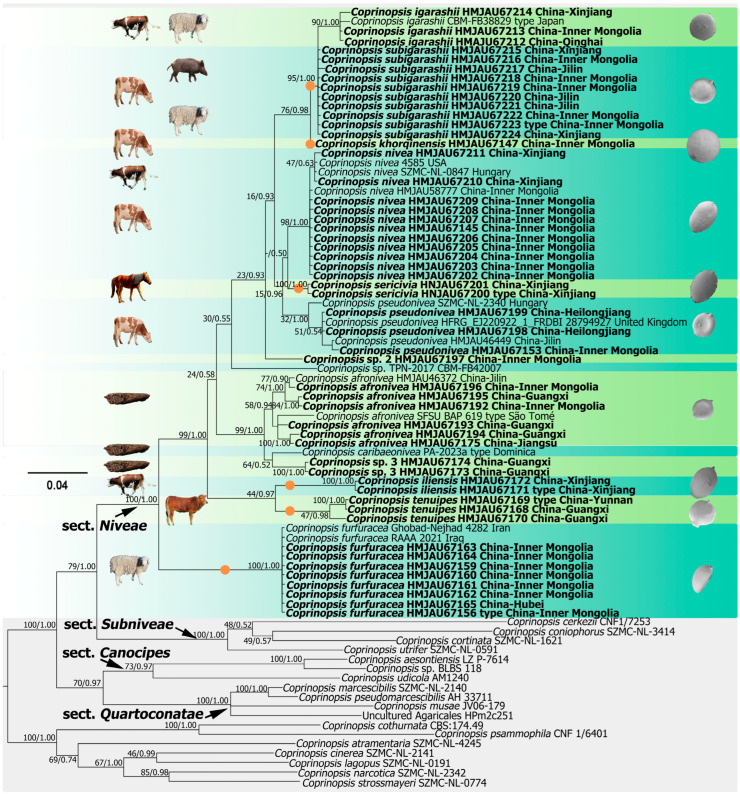
The phylogenetic tree of sect. *Niveae* and related sections of *Coprinopsis* reconstructed with maximum likelihood and Bayesian inference based on a combined ITS-nrLSU-tef-1α-mtSSU dataset. The topology was generated from Bayesian inference; the support value of each node is presented as bootstrap values/Bayesian posterior probabilities. The newly generated sequences from this study are shown in bold font. The proposed new species are highlighted with orange circles.

**Figure 2 jof-10-00835-f002:**
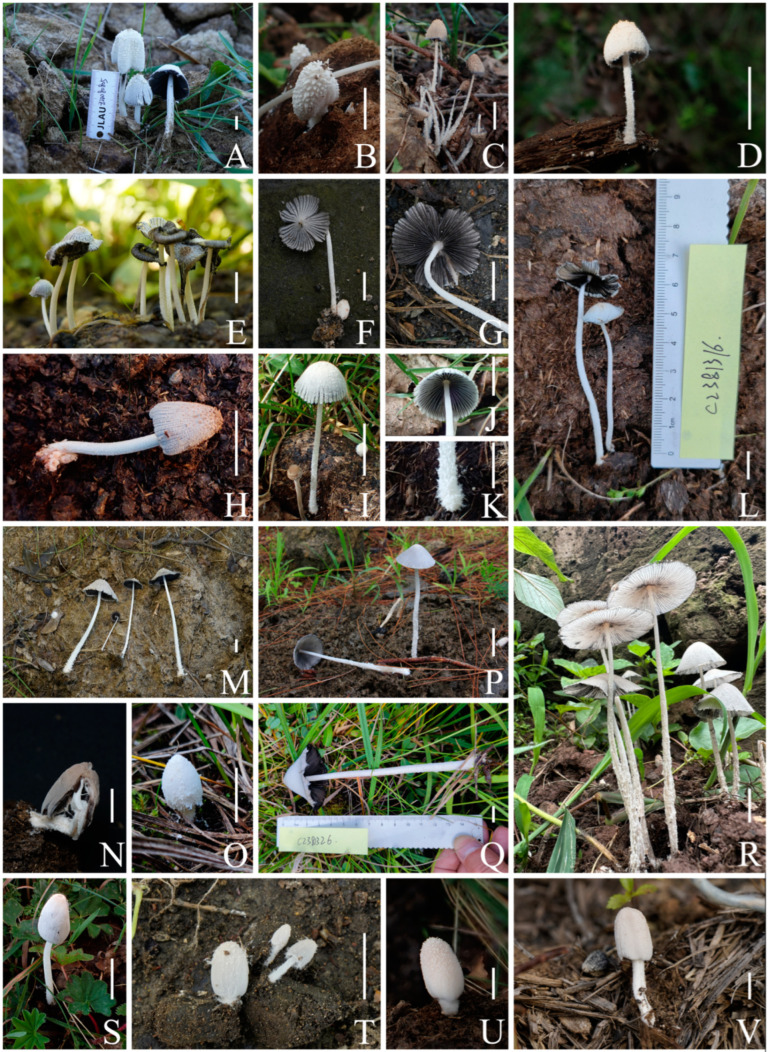
Basidiomata of species of sect. *Niveae* from China. (**A**,**B**) *C. nivea* ((**A**) HMJAU67202; (**B**) HMJAU67145). (**C**,**D**) *C. afronivea* ((**C**) HMJAU67175; (**D**) HMJAU67192). (**E**–**H**) *C. furfuracea* ((**E**) HMJAU67156, type; (**F**,**G**) HMJAU67165; (**H**) HMJAU67159). (**I**–**L**) *C. sericivia* ((**I**–**K**) HMJAU67200, type; **L** HMJAU67201). (**M**,**N**) *C. pseudonivea* ((**M**) HMJAU67198; (**N**) HMJAU67153). (**O**,**Q**) *C. iliensis* ((**O**) HMJAU67172; (**Q**) HMJAU67171, type). (**P**,**R**) *C. tenuipes* ((**P**) KUN-HKAS131920; (**R**) HMJAU67169, type). (**S**) *C. igarashii* (HMJAU67214). (**T**) *C. subigarashii* (HMJAU67223, type). (**U**) *C.* sp. 2 (HMJAU67197). (**V**) *C. khorqinensis* (HMJAU67147, type). Bars: (**A**–**V**) = 1 cm.

**Figure 3 jof-10-00835-f003:**
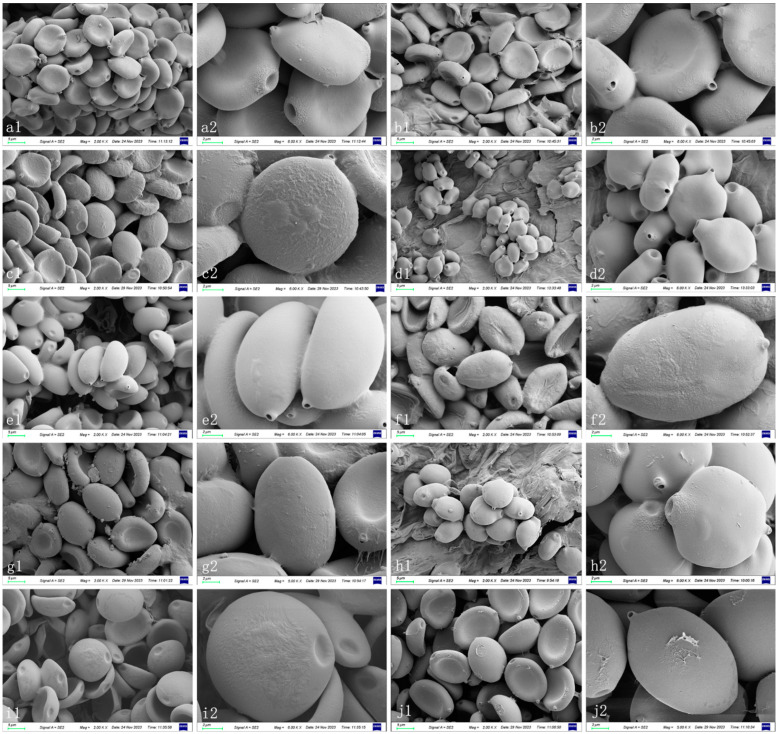
Basidiospores of species of sect. *Niveae*. (**a1**,**a2**) *C. pseudonivea* (HMJAU67153). (**b1**,**b2**) *C. subigarashii*. (**c1**,**c2**) *C. igarashii* (HMJAU67214). (**d1**,**d2**) *C. afronivea* (HMJAU67192). (**e1**,**e2**) *C. furfuracea* (HMJAU67156, type). (**f1**,**f2**) *C. nivea* (HMJAU67219). (**g1**,**g2**) *C. sericivia* (HMJAU67200, type). (**h1**,**h2**) *C. tenuipes* (HMJAU67169, type). (**i1**,**i2**) *C. khorqinensis* (HMJAU67147, type). (**j1**,**j2**) *C. iliensis* (HMJAU67171, type). Bars: (**a1**–**j1**)= 5 μm, (**a2**–**j2**) = 2 μm.

**Figure 4 jof-10-00835-f004:**
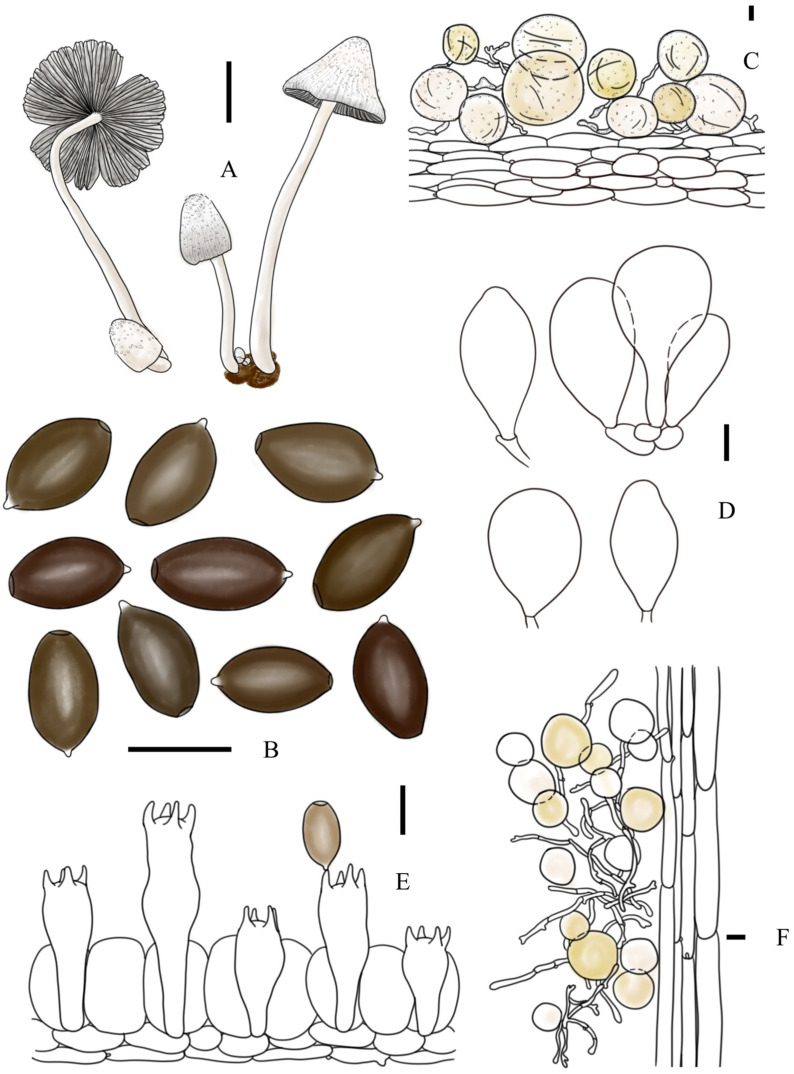
*Coprinopsis furfuracea* (**A**) basidiomata; (**B**) basidiospores; (**C**) radial section of the pileus; (**D**) cheilocystidia; (**E**) basidia and pseudoparaphyses; (**F**) longitudinal section of stipes. Bars: (**A**) = 1 cm; (**B**–**E**) = 10 μm; (**F**) = 20 μm.

**Figure 5 jof-10-00835-f005:**
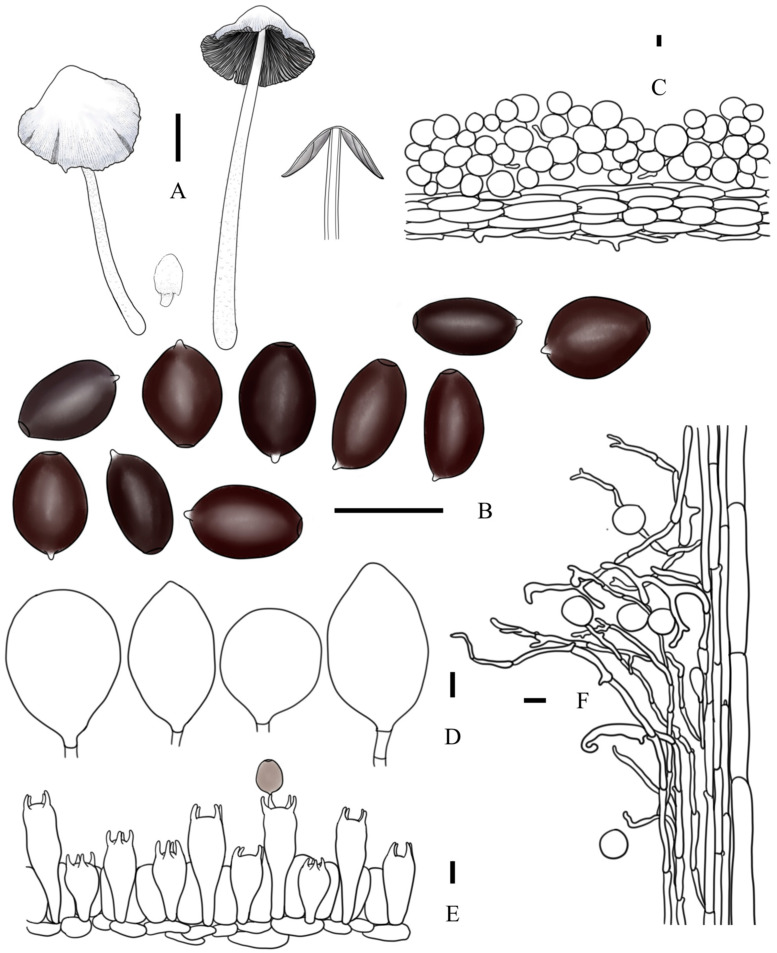
*Coprinopsis iliensis* (**A**) basidiomata; (**B**) basidiospores; (**C**) radial section of pileus; (**D**) cheilocystidia; (**E**) basidia and pseudoparaphyses; (**F**) longitudinal section of stipes. Bars: (**A**) = 1 cm; (**B**–**E**) = 10 μm; (**F**) = 20 μm.

**Figure 6 jof-10-00835-f006:**
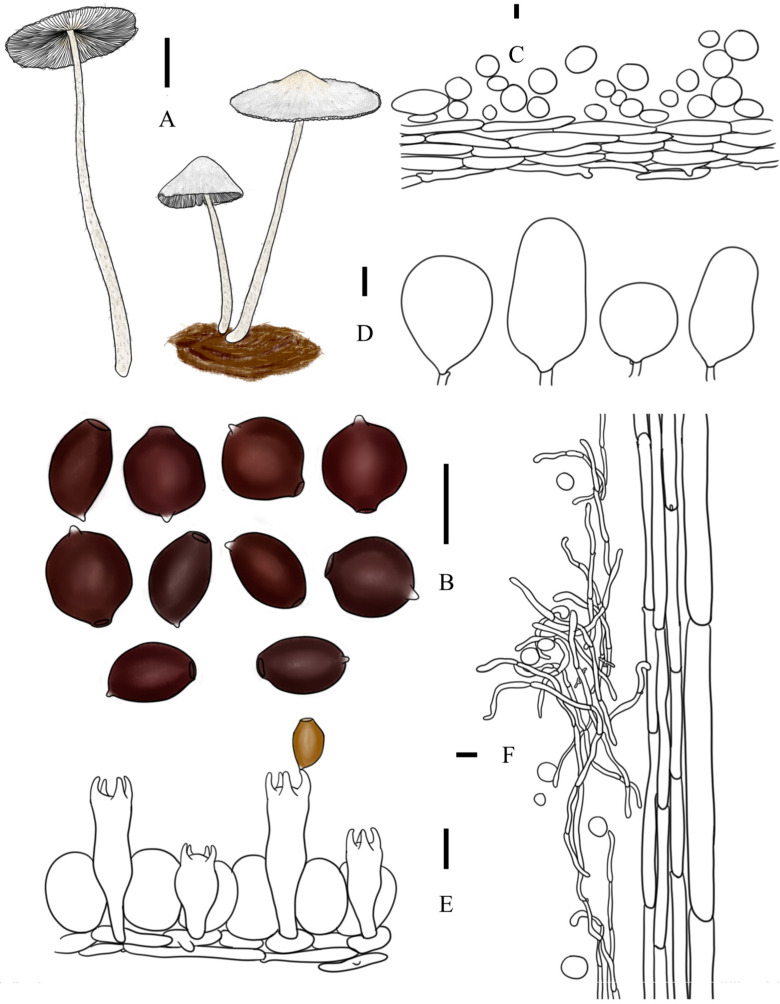
*Coprinopsis tenuipes* (**A**) basidiomata; (**B**) basidiospores; (**C**) radial section of pileus; (**D**) cheilocystidia; (**E**) basidia and pseudoparaphyses; (**F**) longitudinal section of stipes. Bars: (**A**) = 1 cm; (**B**–**E**) = 10 μm; (**F**) = 20 μm.

**Figure 7 jof-10-00835-f007:**
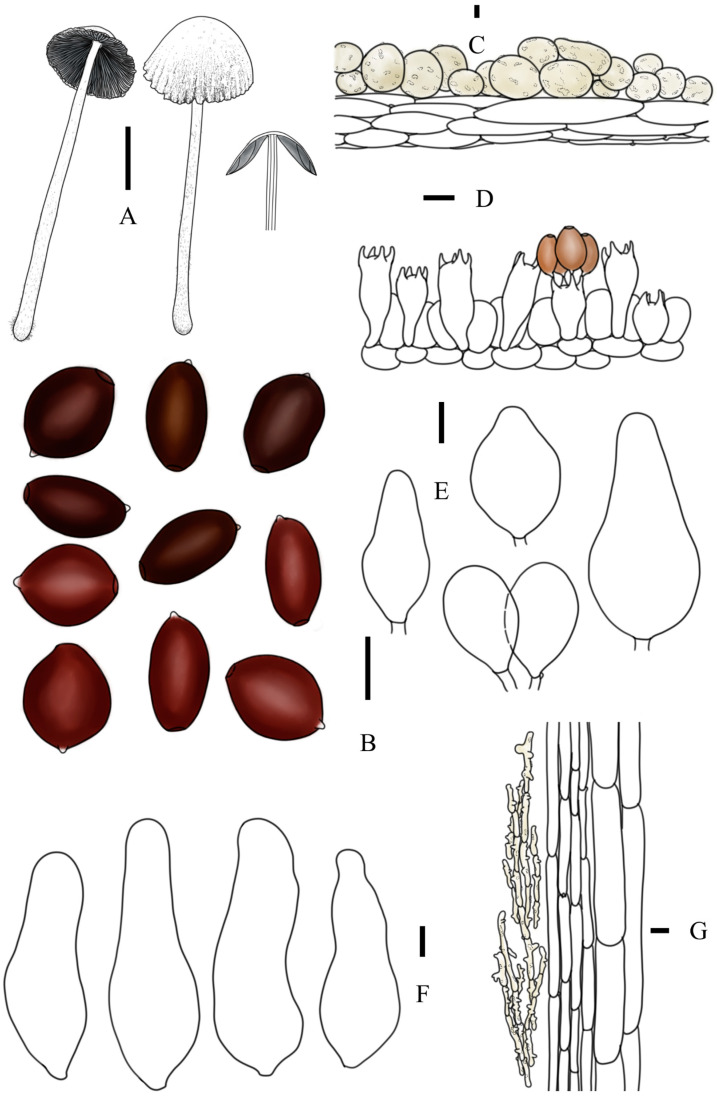
*Coprinopsis sericivia* (**A**) basidiomata; (**B**) basidiospores; (**C**) radial section of pileus; (**D**) basidia and pseudoparaphyses; (**E**) cheilocystidia; (**F**) pleurocystidia; (**G**) longitudinal section of stipes. Bars: (**A**) = 1 cm; (**B**–**F**) = 10 μm; (**G**) = 20 μm.

**Figure 8 jof-10-00835-f008:**
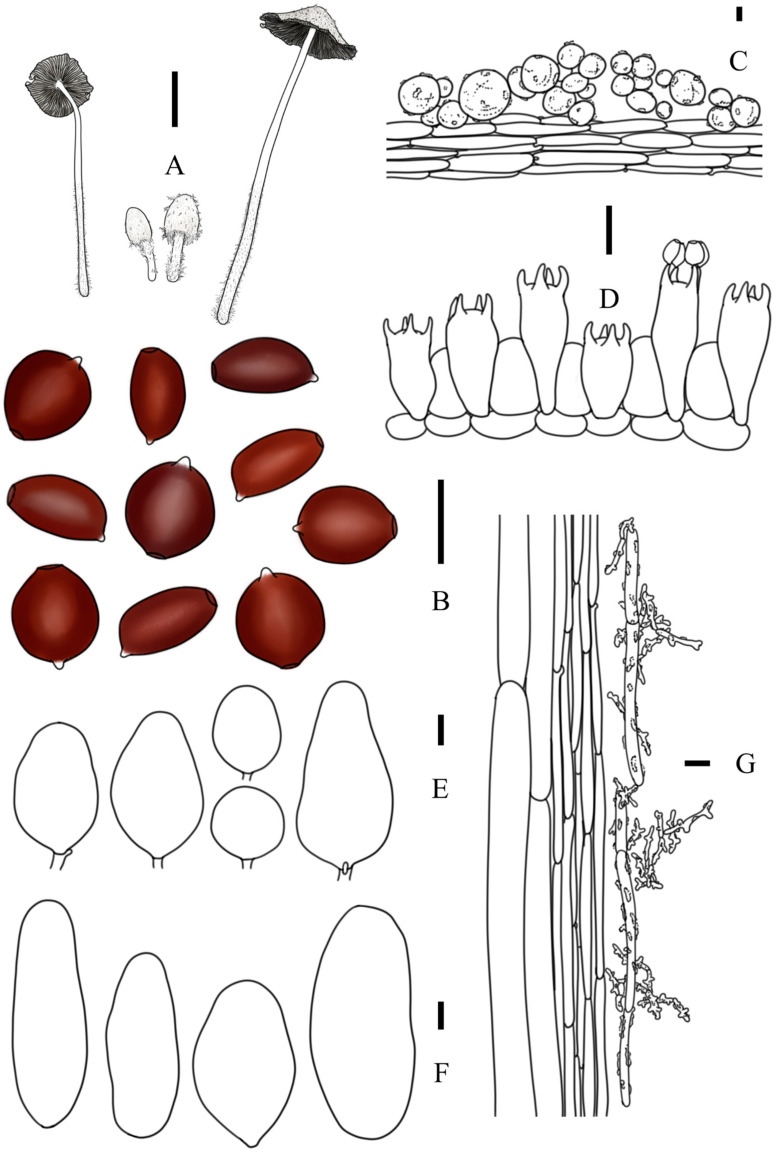
*Coprinopsis subigarashii* (**A**) basidiomata; (**B**) basidiospores; (**C**) radial section of pileus; (**D**) basidia and pseudoparaphyses; (**E**) cheilocystidia; (**F**) pleurocystidia; (**G**) longitudinal section of stipes. Bars: (**A**) = 1 cm; (**B**–**F**) = 10 μm; (**G**) = 20 μm.

**Figure 9 jof-10-00835-f009:**
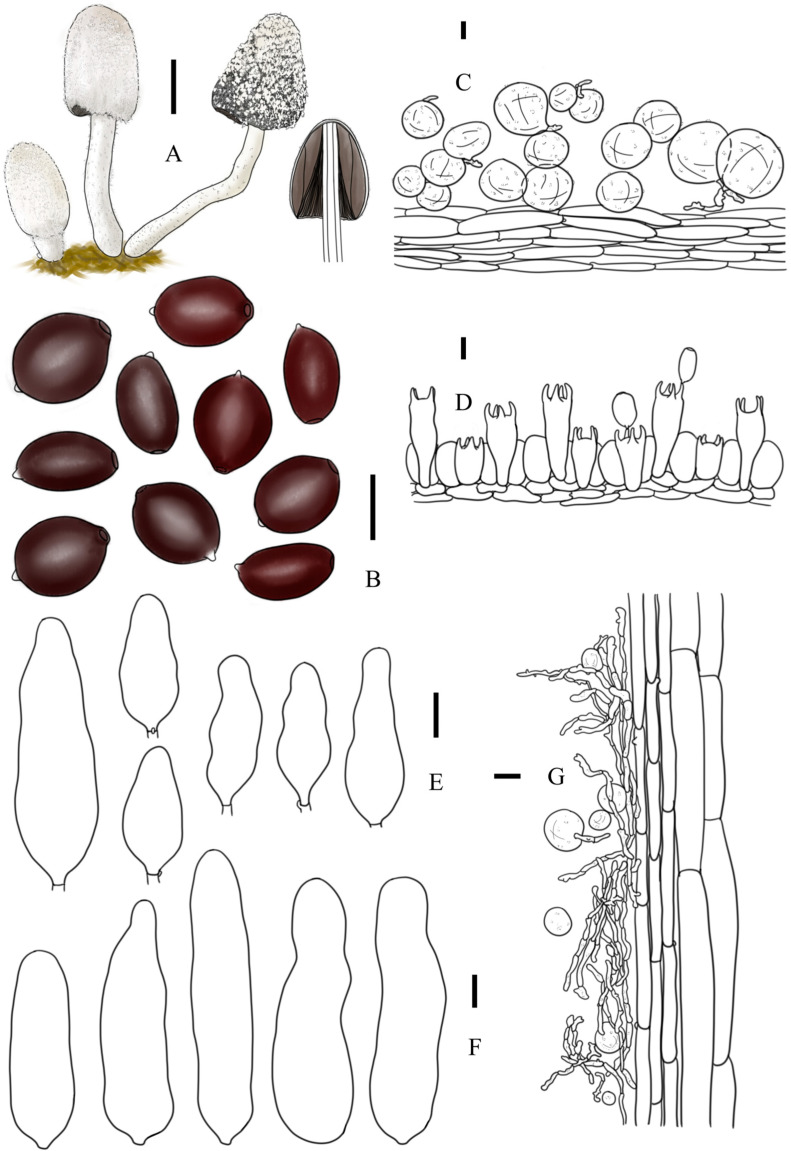
*Coprinopsis khorqinensis* (**A**) basidiomata; (**B**) basidiospores; (**C**) radial section of pileus; (**D**) basidia and pseudoparaphyses; (**E**) cheilocystidia; (**F**) pleurocystidia; (**G**) longitudinal section of stipes. Bars: (**A**) = 1 cm; (**B**–**F**) = 10 μm; (**G**) = 20 μm.

## Data Availability

In publicly available datasets that were analyzed in this study; these data can be found at [https://www.ncbi.nlm.nih.gov/genbank; https://www.mycobank.org/; Zenodo (https://zenodo.org/records/14244210), accessed on 25 November 2024].

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
