# Peer review of "Biodiversity of Herbivores Triggers Species Differentiation of Coprophilous Fungi: A Case Study of Snow Inkcap (Coprinopsis sect. Niveae)"

_jof, 2024, doi:10.3390/jof10120835_

Round 1

Reviewer 1 Report

The manuscript is interesting and enjoyable to read. However, the authors must carefully revise it. Several species names are not in italics, which is unsuitable for a taxonomical study for a taxonomical study. Also, revise the English language and style of tables, key identification, and other details to improve the quality of the manuscript.

Line 27: change “axonomy” by Taxonomy

Line 154 and 155: “devided” change to “divided”

Line 169: “amplying” change to “amplifying”

Lines 196, 197, 198: “C. strossmayeri (Schulzer) Redhead, Vilgalys & 196 Moncalvo, C. narcotica, C. lagopus (Fr.) Redhead, Vilgalys & Moncalvo, C. ci- 197 nerea (Schaeff.) Redhead, Vilgalys & Moncalvo, C. atramentaria, C. pammophila, C. co- 198 thurnata were selected as outgroups based on recent studies.”

The authors must put the name of the species in italic

Line 257: “alomost" change to “almost”

Line 308: “Coprinopsis iliensis T. Bau & L.Y. Zhu sp. Nov” respect the pattern of ICN for Plants and Fungi

Line 380: “strigma” change to “sterigmata”

Line 394: “Apirl" change to "April”

Line 461: C. igarashii write the name on italic

Line 484: “strigma” change to “sterigmata”

573: 3’ “Basidiomata middle- to large-sized, diameter of pileus usually large than 1.5 cm.5.” The authors should improve the style of this key entry

Basidiomata middle- to large-sized, diameter of pileus usually large than 1.5 cm……...5.

Line 584:  “5’ Pileus grey-brown.”8 The authors should improve the style of this key entry

Lines 582-584: Sometimes colour is variable in mushrooms; if the authors had other characters to include in this entry, it would be more accessible to people who study the genus. Please, consider this carefully.

Line 600: C. pseu. See comments of name of species.

Author Response

Comments 1:

Line 257: “alomost" change to “almost”

Line 308: “Coprinopsis iliensis T. Bau & L.Y. Zhu sp. Nov” respect the pattern of ICN for Plants and Fungi

Line 380: “strigma” change to “sterigmata”

Line 394: “Apirl" change to "April”

Line 461: C. igarashii write the name on italic

Line 484: “strigma” change to “sterigmata”

Line 600: C. pseu. See comments of name of species.

Response 1: Thank you for pointing out these mistakes, and we have revised the above-mentioned mistakes in manuscript and marked with highlight.

Comments 2:

573: 3’ “Basidiomata middle- to large-sized, diameter of pileus usually large than 1.5 cm.5.” The authors should improve the style of this key entry

Basidiomata middle- to large-sized, diameter of pileus usually large than 1.5 cm……...5.

Line 584:  “5’ Pileus grey-brown.”8 The authors should improve the style of this key entry

Response 2: The style of the key entry was revised in the newly manuscript.

Comments 3:

Lines 582-584: Sometimes colour is variable in mushrooms; if the authors had other characters to include in this entry, it would be more accessible to people who study the genus. Please, consider this carefully.

Response 3: We agree with this comment. Therefore, we have carefully checked the pileus color of species in sect. Niveae when mature, and confirmed that Coprinopsis nivea, C. iliensis and C. sericivia are snowy white until at aged, while other species are more or less in greyish white or light pinkish brown. Thus, the pileus color was still chosen as distinguishing feature in this section.

Reviewer 2 Report

This paper describes a number of new species in Coprinopsis sect. Niveae in China.

The title suggests that the species diversity and the diversity of herbivores are linked, but nowhere in the text a proper analysis of this is presented, and the preference on dung of various herbivores is nowhere explicitly studied. 

The introduction should first of all clearly state the morphological characters of section Niveae - the authors promise a  description of sect. NIveae, and a discussion of the differences between sect. Niveae and sect. Subniveae, but unfortunately i could not find either in the text. 

Discussion of the evolutionary trends withint the section was also not included; these are subjects for the discussion.

it is interesting to read about the three different categories of dung fungi, recognized by Lundquist - it would be great to see every species that is described in this paper to be designated to one of these categories.

I quit reading this paper halfway as the English needs a total re-write, and for that reason i reject it.

Sentences are un-understandable; this starts already in the abstract, and the first sentence of the introduction.

Terms are mis-spelled throughout (e.g. strigma; instruction instead of incrustation); and many mistakes are re-occurring, and i have not indicated them in every place.

Line 42-43 - the terms used by Persoon are still used today: campanulate for the pileus, and furfuraceous-tomentose for the stipe.

Line 145- ozonium is not present in these species, so delete.

Check that all scientific names are in italics.

Coprinopsis furfuracea - as the name refers to the structure of the pileus, i would expect it to be distinct, but fig. 4 hardly shows it.

When clamp connections are said to be rare, please indicate where they can be found.

Be explicit in the discussions; when using words like ‘this species’ give the name.

It is always very disconcerting to see that the authors of a new species cannot spell it the same way throughout a paper. Here i see khorchinensis, and khorqinensis - what is correct?

In the drawings, C is said to depict a longitudinal section through the pileus. It shows a radial section of the pileipellis with veil elements. 

The key is for species in China only, and that should be made clear. Lay-out of the key needs to be fixed.

References: 

  1. use the Latin name for Leipzig and ‘German’

58. the reference is  “History, morphology and ecology of the Aurochs (Bos taurus primigenius). Lutra 2002, 45, 3–17. 

Author Response

Thank you for the review. All the question you proposed I have revised in the manucript.  

Comments 1:  

Sentences are un-understandable; this starts already in the abstract, and the first sentence of the introduction.

Terms are mis-spelled throughout (e.g. strigma; instruction instead of incrustation); and many mistakes are re-occurring, and i have not indicated them in every place.

Response 1:  This problem has been solved in the article.

Comments 2: 

Line 42-43 - the terms used by Persoon are still used today: campanulate for the pileus, and furfuraceous-tomentose for the stipe.

Line 145- ozonium is not present in these species, so delete.

Response 2: The problem has been solve in the manucript.

Comments 3: Coprinopsis furfuracea - as the name refers to the structure of the pileus, i would expect it to be distinct, but fig. 4 hardly shows it.

Respond 3: The illustration of Coprinopsis furfuracea has been revised.

Comments 4: It is always very disconcerting to see that the authors of a new species cannot spell it the same way throughout a paper. Here i see khorchinensis, and khorqinensis - what is correct?

Response 4: It is "khorqinensis", and the full text has been modified.

Comments 5: In the drawings, C is said to depict a longitudinal section through the pileus. It shows a radial section of the pileipellis with veil elements. 

Response 5: All the C are change to "radial section".

Comments 6: The key is for species in China only, and that should be made clear. Lay-out of the key needs to be fixed.

Response 6: It is coverd species only China but also the world. The text has been fixed.

Comments 7: 

  1. use the Latin name for Leipzig and ‘German’

58. the reference is  “History, morphology and ecology of the Aurochs (Bos taurus primigenius). Lutra 2002, 45, 3–17. 

Response 7: These have been revised in manucript.

Reviewer 3 Report

A global study of the fungi Coprinopsis sect. Niveae was conducted. It was based on a large amount of factual material. The authors collected a lot of field samples in China with a detailed record of the distribution and the animal origin of the dung. In addition, a full set of data for this group from the GenBank database was used for phylogenetic analysis. Such a large-scale work allowed the authors to expand the section to 14 phylogenetic species, 6 of which are new to science.

In this study, detailed research of the macro-, micro- and ultramicro- morphology of the species collected from China was carried out in combination with a four-loci phylogenetic studies of Coprinopsis sect. Niveae. The authors provided detailed descriptions of new species, high-quality drawings and photographs. The work complies with all formal requirements for the description of new species, all newly generated sequences were deposited in GenBank, new names were deposited in Mycobank. It is significantly that the features important for identification have been established, among which the origin of the substrate is remarkable. The given key is very important for identifying the species of this section.

However, there are some shortcomings in this paper.

The Abstract states that the paper describes (line 18) «six novel species, namely Coprinopsis furfuracea, C. iliensis, C. khorqinensis, C. sericivia, C. subigarashii, and C. tenuipes». Further in the Introduction (line 124) «here we will discuss: 1) four new species of this section». In reality, this paper describes six new species.

In Figure 1 «The proposed new species were highlighted with orange circles». Only five species are highlighted (C. khorqinensis is missing).

The Key to species belonging of sect. Niveae includes only nine species. Coprinopsis subigarashii and C. sericivia, described in this paper, are missing. These need to be added to the key to enable identification.

 The Introduction states (line 126–127) that the paper discusses «3) the description of sect. Niveae; 4) the difference between sect. Niveae and sect. Subniveae.». However, this information is missing from this paper.

 The discussion is poorly structured and would benefit from a reworking.

 Incorrect use of terms:

«Basidia trimorphological» (line 267, 334, 379, 542) or «Basidia dimorphological» (line 431, 483). It is unclear what is meant by this. If there are really three or two types of basidia, then they should all be described with all detail. In the figures provided, the basidia do not differ in structure and shape, only in size, which corresponds to different stages of their formation. The basidia elongate as the basidiospores mature.

"pseudoparaphyses" (line 269, 336, 380, 433, 485, 543) is an out-of-use term, correctly basidioles or probasidia (an immature or aborted basidium).

There are typos in the work:

line 2 – "axonomy;" – Taxonomy.

line 32 – "although it was fracastorius [1]" – Fracastorius.

line 263, 330, 374, 425, 477, 537 – "Smell somewhat like rotten grass, order unknown." – oder.

line 268, 335, 380, 432, 484, 543 – "4-spored, strigma 3−8" – sterigma.

Author Response

Comments 1: The Abstract states that the paper describes (line 18) «six novel species, namely Coprinopsis furfuracea, C. iliensis, C. khorqinensis, C. sericivia, C. subigarashii, and C. tenuipes». Further in the Introduction (line 124) «here we will discuss: 1) four new species of this section». In reality, this paper describes six new species.

Response 1: The article has been revised.

Comments 2: In Figure 1 «The proposed new species were highlighted with orange circles». Only five species are highlighted (C. khorqinensis is missing).

Response 2: The article has been revised.

Comment 3: The Key to species belonging of sect. Niveae includes only nine species. Coprinopsis subigarashii and C. sericivia, described in this paper, are missing. These need to be added to the key to enable identification.

Response 3: The key has been revised.

Comments 4: The Introduction states (line 126–127) that the paper discusses «3) the description of sect. Niveae; 4) the difference between sect. Niveae and sect. Subniveae.». However, this information is missing from this paper.

Response 4: Those parts have been added.

Comments 5: The discussion is poorly structured and would benefit from a reworking.

Response 5: The discussion has been rewritten now.

Comments 5: «Basidia trimorphological» (line 267, 334, 379, 542) or «Basidia dimorphological» (line 431, 483). It is unclear what is meant by this. If there are really three or two types of basidia, then they should all be described with all detail. In the figures provided, the basidia do not differ in structure and shape, only in size, which corresponds to different stages of their formation. The basidia elongate as the basidiospores mature.

Responds 5: The basidia are in mature status and the illustration and description have been changed.

Comments 6: "pseudoparaphyses" (line 269, 336, 380, 433, 485, 543) is an out-of-use term, correctly basidioles or probasidia (an immature or aborted basidium).

Respond 6: “pseudoparaphyses” is a word usually used in coprinoid fungi and is still used in year 2023, like in Coprinopsis caesia and C. fulva, the newest species in Coprinopsis. Therefore, we used it for misunderstanding.

Comments 7: There are typos in the work:

line 2 – "axonomy;" – Taxonomy.

line 32 – "although it was fracastorius [1]" – Fracastorius.

line 263, 330, 374, 425, 477, 537 – "Smell somewhat like rotten grass, order unknown." – oder.

line 268, 335, 380, 432, 484, 543 – "4-spored, strigma 3−8" – sterigma.

Response 7: Thank you for pointing out these problems, we have modified in the text.

Reviewer 4 Report

The authors have done a great deal of interesting work, but unfortunately, they were not always careful.

Detail comments are given in the text.

The authors have done a great deal of interesting work, but unfortunately, they were not always careful.

First of all – in the nomenclature part. The name Coprinus is masculine. Then the section name in the plural is Nivei. The name Coprinopsis is feminine. Then the plural is the section name – Niveae.

Art. 21.2. The epithet in the name of a subdivision of a genus is either of the same form as a generic name, or a noun in the genitive plural, or a plural adjective agreeing in gender with the generic name…

This must be checked throughout the text. It is also necessary to check the authorship of generic and species names.

In many places, italics of species names are missing.

At the beginning of a new paragraph, the generic name must be written in full.

The identification key must be formatted carefully.

The DOI are absent in the reference list.

Specific comments are given in the text.

Author Response

Thanks for your kind review, and I have revised in the manucrift.

Comments 1: First of all – in the nomenclature part. The name Coprinus is masculine. Then the section name in the plural is Nivei. The name Coprinopsis is feminine. Then the plural is the section name – Niveae.

Response 1: This problem has been solved in article.

Comments 2:  

This must be checked throughout the text. It is also necessary to check the authorship of generic and species names.

In many places, italics of species names are missing.

At the beginning of a new paragraph, the generic name must be written in full.

Response 2: Thank you for the kind reminds and I've checked and revised it.

Comments 3: The identification key must be formatted carefully.

Response 3:  The identification key has been formatted.

Comments 4: The DOI are absent in the reference list.

Response 4: The DOI is given if it present.

Round 2

Reviewer 2 Report

see my earlier comments

the English needs a native English speaker for editing and corrections. i stopped halfway. It was one of my main complaints for the first review, but still stands. I am not a native English speaker, nor an editor for this journal. This is the responsibility of the authors to present a readable, understandable and accurate english text.

Author Response

Thank you for your review. Based on your comments, we have made detailed revisions, which can be found in the sections with a green background in the text.